# KIFC1 Is Associated with Basal Type, Cisplatin Resistance, PD-L1 Expression and Poor Prognosis in Bladder Cancer

**DOI:** 10.3390/jcm10214837

**Published:** 2021-10-21

**Authors:** Yohei Sekino, Quoc Thang Pham, Kohei Kobatake, Hiroyuki Kitano, Kenichiro Ikeda, Keisuke Goto, Tetsutaro Hayashi, Hikaru Nakahara, Kazuhiro Sentani, Naohide Oue, Wataru Yasui, Jun Teishima, Nobuyuki Hinata

**Affiliations:** 1Department of Urology, Graduate School of Biomedical and Health Sciences, Hiroshima University, Hiroshima 734-8551, Japan; koukoba2710@gmail.com (K.K.); tanokin@hiroshima-u.ac.jp (H.K.); kenikeda@hiroshima-u.ac.jp (K.I.); keigoto@hiroshima-u.ac.jp (K.G.); tetsu-haya@hiroshima-u.ac.jp (T.H.); teishima@hiroshima-u.ac.jp (J.T.); hntnbyk@gmail.com (N.H.); 2Department of Molecular Pathology, Graduate School of Biomedical and Health Sciences, Hiroshima University, Hiroshima 734-8551, Japan; quocthang388@gmail.com (Q.T.P.); kzsentani@hiroshima-u.ac.jp (K.S.); naoue@hiroshima-u.ac.jp (N.O.); wyasui@hiroshima-u.ac.jp (W.Y.); 3Department of Gastroenterology and Metabolism, Graduate School of Biomedical and Health Sciences, Hiroshima University, Hiroshima 734-8551, Japan; hnkhr@hiroshima-u.ac.jp

**Keywords:** bladder cancer, *KIFC1*, basal type, p53, cisplatin, PD-L1

## Abstract

Kinesin family member C1 (*KIFC1*), a minus end-directed motor protein, is reported to play an essential role in cancer. This study aimed to analyze *KIFC1* expression and examine *KIFC1* involvement in cisplatin resistance in bladder cancer (BC). Immunohistochemistry showed that 37 of 78 (47.4%) BC cases were positive for *KIFC1*. *KIFC1*-positive cases were associated with high T stage and lymph node metastasis. Kaplan-Meier analysis showed that *KIFC1*-positive cases were associated with poor prognosis, consistent with the results from public databases. Molecular classification in several public databases indicated that *KIFC1* expression was increased in basal type BC. Immunohistochemistry showed that *KIFC1*-positive cases were associated with basal markers 34βE12, CK5 and CD44. *KIFC1* expression was increased in altered *TP53* compared to that in wild-type *TP53*. Immunohistochemistry showed that *KIFC1*-positive cases were associated with p53-positive cases. P53 knockout by CRISPR-Cas9 induced *KIFC1* expression in BC cell lines. Knockdown of *KIFC1* by siRNA increased the sensitivity to cisplatin in BC cells. Kaplan-Meier analysis indicated that prognosis was poor among *KIFC1*-positive BC patients treated with cisplatin-based chemotherapy. Immunohistochemistry showed that *KIFC1*-positive cases were associated with PD-L1-positive cases. High *KIFC1* expression was associated with a favorable prognosis in patients treated with atezolizumab from the IMvigor 210 study. These results suggest that *KIFC1* might be a promising biomarker and therapeutic target in BC.

## 1. Introduction

Bladder cancer (BC) is the 11th most commonly diagnosed cancer worldwide, with approximately 573,000 new cases and 213,000 deaths in 2020 [1]. BC can be classified into two types: non-muscle-invasive BC and muscle-invasive BC (MIBC). In non-MIBC, T1 tumors are an aggressive subtype with 40% recurrence and 15% progression to MIBC at 5 years [2]. MIBC will eventually develop distant metastasis resulting in a 5 year survival rate of <50% [3]. Although standard care for MIBC is neoadjuvant chemotherapy followed by radical cystectomy, about 40% of patients experience relapse [4]. Cisplatin-based chemotherapy is the standard first-line treatment for patients with relapse after radical cystectomy [3]. However, most patients receive few benefits due to cisplatin resistance. Therefore, clarifying the molecular biology of cancer progression and cisplatin resistance is urgently needed in BC.

The presence of more than two centrosomes (centrosome amplification: CA) affects the chromosome segregation machinery and leads to chromosomal instability [5]. Several reports have shown that CA correlates with aggressive features and poor prognosis in BC [6,7]. Although CA causes multipolar spindles and leads to apoptosis, cancer cells overcome these lethal effects through centrosome clustering. Centrosome clustering, defined as the reshaping of transient multipolar spindles into pseudo-bipolar structures, is a well-studied mechanism that allows cancer cells to avoid apoptosis [8]. Kinesin family member C1 encoded by the *KIFC1* gene (also called *HSET*) belongs to the kinesin family member of motor proteins and is implicated in centrosome clustering, microtubule transport and spindle formations during mitosis [9]. A recent study showed that *KIFC1* promoted cell growth and epithelial-mesenchymal transition in BC [10]. However, the biological role of *KIFC1* in BC has not been fully elucidated.

In this study, we performed immunohistochemistry to analyze the prognostic value of *KIFC1* and examined the association between *KIFC1* and CD44, CK5, 34βE12, p53 and PD-L1 in BC. We also investigated the association between *KIFC1* and molecular classification, analyzed the role of *KIFC1* in cisplatin resistance, and performed in silico analysis of the role of *KIFC1* in immunotherapy.

## 2. Materials and Methods

### 2.1. Tissue Samples

In total, 174 tumors were used in this retrospective study, of which 78 tumors were collected from patients diagnosed as having BC who underwent cystectomy at Hiroshima University Hospital (Hiroshima, Japan) (Appendix A) and 50 tumors were collected from patients diagnosed as having BC who underwent cystectomy at Kure Medical Center and Chugoku Cancer Center (Kure, Japan) (Appendix A). In addition, 46 tumors were collected from patients diagnosed as having BC treated with cisplatin-based chemotherapy at Hiroshima University Hospital (Hiroshima, Japan). The Institutional Review Boards of both institutions approved this study (Hiroshima University, IRB# E912; Kure Medical Center/Chugoku Cancer Center: 2019-08).

### 2.2. Immunohistochemistry

Immunohistochemistry was performed as described previously [11]. We used archival formalin-fixed, paraffin-embedded tissues from the 174 patients with BC for immunohistochemical analysis. Tumor staging was performed according to the TNM (tumor-node-metastasis) classification system [12]. Sections were incubated with anti-*KIFC1* antibody (1:100, H00003833-M01, Abnova, Taipei, Taiwan), CD44 (1:200, M7082, Dako, Glastrup, Denmark, USA), CK5 (1:200, M7237, Dako, Glastrup, Denmark), 34βE12 (1:200, GA051, Dako, Glastrup, Denmark), Ki-67 (1:100, M7240, Dako, Glastrup, Denmark), p53 (1:200, M7001, Dako, Glastrup, Denmark) and PD-L1 (1:300, ab205921, Abcam, MA, USA) for 1 h at room temperature. *KIFC1* expression in BC was scored in all tumors as positive or negative. When more than 10% of tumor cells were stained, the specimen was considered positive for *KIFC1* (according to the median cut-off values rounded off to the nearest 10%). The expressions of CD44, CK5, 34βE12, Ki-67, p53 and PD-L1 were also scored in all tumors as positive or negative. When more than 10% of tumor cells showed staining, the immunostaining of CD44, CK5, 34βE12 was considered positive. When more than 20% of tumor cells showed staining, the immunostaining of Ki-67 was considered positive. p53 staining was evaluated based on the study [13]. Immunostaining of PD-L1 was considered positive according to median cutoff values rounded off to the nearest 5%. Using these definitions, two observers (K.S. and N.O.) without knowledge of the patients’ clinical and pathologic parameters or outcomes independently reviewed immunoreactivity in each specimen.

### 2.3. In Silico Analysis

The GEPIA web tool was used to determine *KIFC1* expression in The Cancer Genome Atlas (TCGA) (BLCA) dataset [14]. The expression array data were downloaded from GEO and Array Express under accession numbers GSE120736 [15], GSE13507 [16], GSE32548 [17], GSE48277 [18], GSE124305 [19], GSE154261 [20], E-MTAB-1803 [21] and E-MTAB-4321 [22]. The data from the study by Sanchez et al. [23] and that from the study by Taber et al. [24] were downloaded. Clinicopathologic characteristics of bladder cancer patients from GSE13507, GSE32548 and GSE48277 (Appendix A). The data from the IMvigor 210 study was also downloaded from Roche, MA, USA., Data signature analysis was performed with the UCSC web tool [25]. The proliferation signature was referred to the study by Tuan et al. [26] (Appendix A).

### 2.4. Cell Lines

Four cell lines derived from human BC (RT4, RT112, 5637 and UMUC3) were provided by the Vancouver Prostate Centre (Vancouver, BC, Canada). The cells were maintained in RPMI 1640 (Nissui Pharmaceutical Co., Ltd., Osaka, Japan) containing 10% fetal bovine serum (BioWhittaker, Walkersville, MD, USA) in a humidified atmosphere with 5% CO_2_ at 37 °C.

### 2.5. Western Blotting

Western blotting was performed as described previously [27]. Lysates were solubilized in Laemmli sample buffer by boiling and subjected to 10% SDS-polyacrylamide gel electrophoresis followed by electro-transfer onto a nitrocellulose filter. The membrane was incubated with a primary antibody for *KIFC1* (1:500, H00003833-M01, Abnova, Taipei, Taiwan), CD44 (1:1000), and p53 (DO-1) (1:1000, Cell Signaling Technology, Inc., Danvers, MA, USA). Peroxidase-conjugated anti-mouse IgG or anti-rabbit IgG was used in the secondary reaction. Immunocomplexes were visualized with an ECL Western Blot Detection System (Amersham Biosciences, Piscataway, NJ, USA). β-Actin (Sigma-Aldrich, St. Louis, MO, USA) was also stained as a loading control.

### 2.6. Generation of p53 Knockout Cells

To knock out p53 in RT4 and RT112 cells, we used CRISPR-Cas9 technology, which was performed as described previously [28]. p53 single-guide RNAs (sgRNAs; CRISPR-P53 vector) and scrambled sgRNAs (empty vector) were purchased from ABM Inc. (Richmond, BC, Canada). The sgRNA sequence of the CRISPR-P53 vector was GACGGAAACCGTAGCTGCCC. Lentiviral particles were generated by co-transfection of HEK 293T cells with Cas9-sgRNA constructs and packaging plasmids (GAG, VSVG and REV). After 48 h, the conditioned media containing lentiviral particles were harvested and used to infect cells using Polybrene as the transfection agent. Stable p53 knockout cells were selected by passaging in media containing 4 µg/mL puromycin.

### 2.7. Cisplatin Treatment

Cisplatin treatment was performed as described previously [29]. Cisplatin (Nippon Kayaku Co., Ltd., Tokyo, Japan) was obtained and handled according to the manufacturer’s recommendations. Cell lines treated with vehicle (0.5% ethanol) or escalating doses of cisplatin were assessed for cell viability. A WST-1 assay was performed at 48 h after cisplatin chemotherapy [15]. Drug sensitivity curves and IC_50_ values were calculated using GraphPad Prism 4.0 software (GraphPad Software Inc., San Diego, CA, USA).

### 2.8. Statistical Analysis

All experiments were repeated at least three times with each sample in triplicate. The results are expressed as the mean ± SD of the triplicate measurements. Sample sizes for relevant experiments were determined by power analysis. Statistical differences were evaluated using the two-tailed Student *t*-test or Mann-Whitney U-test. One-way analysis of variance (ANOVA) was used to determine whether there were any statistically significant differences. A *p*-value of <0.05 was considered statistically significant. After a Kaplan-Meier analysis was performed, any statistical difference between the survival curves of the cohorts was determined with the log-rank Mantel-Cox test. Statistical analyses were conducted primarily using GraphPad Prism software (GraphPad Software Inc., San Diego, CA, USA) or JMP14 (SAS Institute, Cary, NC, USA).

## 3. Results

### 3.1. Expression of KIFC1 in BC

We performed immunohistochemistry to analyze the expression of *KIFC1* in 78 BC tissue samples (Hiroshima cohort, Appendix A). Weak or no staining of *KIFC1* was observed in the non-neoplastic urothelium, whereas stronger and more extensive staining was observed in BC tissues (Figure 1A). Staining of *KIFC1* was mainly observed in the nucleus in BC (Figure 1B). In total, 37 (47.4%) of the BC cases were considered positive for *KIFC1*. These positive cases were associated with high T stage and lymph node metastasis (Table 1). *KIFC1* expression was increased in superficial BC and MIBC compared to that in normal urothelium in the study by Sanchez et al. [21] (Figure 1C). *KIFC1* expression was increased in high T stage cancer in the study GSE120736 (Figure 1D). Of note, high *KIFC1* expression was associated with poor recurrence-free survival among the patients with T1 BC in the study GSE154261 (Figure 1E). High *KIFC1* expression was also associated with poor progression-free survival among the patients with T1 BC in the studies GSE154261 and E-MTAB-4321 (Figure 1F,G). These results indicate that *KIFC1* plays an essential role in progression in BC.

### 3.2. Prognostic Value of KIFC1 after Cystectomy in BC

We next analyzed the prognostic value of *KIFC1* after cystectomy in BC. A Kaplan-Meier analysis showed that the *KIFC1*-positive cases were significantly associated with poor cancer-specific survival (hazard ratio 6.443, *p* < 0.001) and overall survival in BC (hazard ratio 3.159, *p* < 0.001) in the Hiroshima cohort (Figure 2A,B). To verify our findings, we analyzed the prognostic value of *KIFC1* in BC using the public databases. A Kaplan-Meier analysis showed that high *KIFC1* expression was significantly associated with poor prognosis in GSE13507, GSE32548 and GSE48277 (Figure 2C–E). We performed univariate and multivariate Cox proportional hazard analyses to evaluate the potential use of *KIFC1* expression as a prognostic marker. In the multivariate model, positive *KIFC1* expression was independently associated with poor overall survival (hazard ratio 3.121, *p* = 0.009; Table 2).

### 3.3. KIFC1 Is Increased in Basal Type BC

Several recent studies have reported the clinical significance of molecular classifications in BC [30]. Therefore, we analyzed the association between *KIFC1* expression and molecular classifications. In TCGA-BLCA, *KIFC1* expression was higher in basal/squamous and neuronal type BC than that in other BC types (Figure 3A). In the study GSE124305, *KIFC1* expression was higher in basal type BC than that in other BC types (Figure 3B). In the IMvigor 210 study, *KIFC1* expression was higher in basal/squamous and genomically unstable type BC than that in other BC types (Figure 3C). These findings indicate that *KIFC1* expression was increased in basal type BC. Therefore, we performed immunohistochemistry of basal markers (34βE12, CK5, and CD44) in 50 patients with BC from the Kure cohort. Immunohistochemistry showed that positive *KIFC1* cases were associated with positive 34βE12, CK5 and CD44 cases in this cohort (Figure 3D) (Table 3). Of note, western blotting showed that *KIFC1* knockdown suppressed CD44 expression in 5637 and UMUC3 cells (Figure 3E), indicating that *KIFC1* is involved in basal differentiation.

### 3.4. KIFC1 Is Involved in Cell Proliferation in BC

As mentioned in the introduction, *KIFC1* promotes bladder cancer cell proliferation in vitro [10]. Therefore, we validated this finding by immunohistochemistry and signature analysis. Immunohistochemistry showed that positive *KIFC1* cases were associated with positive Ki-67 cases in Hiroshima cohort (Table 4). What is more, *KIFC1* expression was positively correlated with proliferation signature value [26] in TCGA cohort (Figure 4A). Proliferation signature value was increased in basal/squamous type than that in luminal infiltrated and luminal papillary types (Figure 4B).

### 3.5. KIFC1 Is Associated with Genomic Instability in BC

A recent study has shown that *KIFC1* phosphorylation induces chromosomal instability in breast cancer [31]. As shown in Figure 3C, *KIFC1* expression was increased in genomically unstable type BC in the IMvigor 210 study. In the study by Taber et al. [22], *KIFC1* expression was higher in high genomic instability than in low genomic instability groups (Figure 5A). *KIFC1* expression was positively correlated with aneuploidy score in TCGA-BLCA (Figure 5B).

### 3.6. KIFC1 Is Regulated by p53 in BC

A comprehensive sequencing study found that half of the patients with MIBC had a *TP53* mutation [32], indicating p53 pathway plays an essential role in biology of BC. What is more, cancers with a loss of p53 showed increased genomic instability [33]. Therefore, we analyzed the association between *KIFC1* and p53. In TCGA-BLCA, gene alteration of *KIFC1* was associated with gene alteration of *TP53* (Table 5). Furthermore, mRNA *KIFC1* expression was increased in the case of *TP53* alteration in TCGA-BLCA and E-MTAB-1803 (Figure 6A,B). Then, we performed immunohistochemistry of p53 in 58 bladder cancer patients. We considered p53 overexpression and p53 complete absence of expression as altered-type p53 based on the study [13]. Immunohistochemistry showed that positive *KIFC1* cases were associated with altered-type p53 cases in 58 patients with BC from the Hiroshima cohort (Figure 6C) (Table 6). Then, we established p53 knockout cells using CRISPR-Cas9 in RT4 and RT112 cells, which are p53 wild type BC cell lines [34] to analyze the effect of p53 knockout on *KIFC1* expression. Western blotting showed that *KIFC1* expression was upregulated in the *TP53* knockout cells (Figure 6D).

### 3.7. KIFC1 Is Involved in Cisplatin Resistance in BC

Recent studies have shown that *KIFC1* is involved in cisplatin resistance in breast cancer [31,35]. Therefore, we analyzed the involvement of *KIFC* in cisplatin resistance in BC. We performed WST-1 assays to measure cell viability under various concentrations of cisplatin in 5637 and UMUC3 cells transfected with negative control small interfering RNA (siRNA) and KIFC1 siRNA. *KIFC1* knockdown increased the sensitivity to cisplatin in the 5637 and UMUC3 cells (Figure 7A,B). Then, to analyze the prognostic value of *KIFC* for cisplatin treatment, we performed immunohistochemistry of *KIFC* in 46 patients with advanced BC treated with cisplatin-based chemotherapy (Table 7). *KIFC1* positive cases was not associated with the response to cisplatin-based chemotherapy (Table 8). Kaplan-Meier analysis showed that *KIFC1*-positive cases were associated with poor prognosis after cisplatin-based chemotherapy (Figure 7C). These results suggest that *KIFC1* may be a prognostic marker for cisplatin-based chemotherapy.

### 3.8. KIFC1 Is Associated with PD-L1 and Favorable Prognosis after PD-L1 Inhibition in BC

We performed immunohistochemistry of PD-L1 in BC (Figure 8A), which showed that *KIFC1*-positive cases were significantly associated with PD-L1-positive cases (Table 3). PD-L1 is used as a biomarker for immune checkpoint inhibitors (ICI) [36]. Therefore, we analyzed the role of *KIFC1* for ICI. High *KIFC1* expression was significantly associated with the favorable outcome (complete response/partial response) (Table 9). Kaplan-Meier analysis showed that high *KIFC1* expression was significantly associated with favorable overall survival in BC treated with atezolizumab (Figure 8B). These results suggest that *KIFC1* may be a useful marker for atezolizumab treatment.

## 4. Discussion

Molecular mechanisms of cisplatin resistance are believed to be caused by multiple factors including drug uptake and efflux, detoxification, DNA repair and apoptosis [37]. However, strategies to overcome cisplatin resistance are not well established. A recent study showed that *KIFC1* phosphorylation by ATM and ATR kinase is involved in drug resistance in breast cancer [31]. Another study reported that *KIFC1* knockdown increased the sensitivity to cisplatin in breast cancer [35]. In the present study, we showed that knockdown of *KIFC1* increased the sensitivity to cisplatin, which is the first report to analyze the involvement of *KIFC1* in cisplatin resistance in BC. These results suggest that *KIFC1* may play an essential role in cisplatin resistance. In this study, western blotting showed that *KIFC1* knockdown suppressed CD44 expression in BC cell lines. Previously, we showed that *KIFC1* is associated with CD44 in prostate cancer and gastric cancer [38,39]. Several studies have reported that CD44 is involved in cisplatin resistance in BC [40,41], which may help to explain why *KIFC1* increased the sensitivity to cisplatin. Collectively, these results suggest that knockdown of *KIFC1* increased the sensitivity to cisplatin partly through CD44 in BC.

Immunohistochemistry in the present study showed that *KIFC1*-positive cases were associated with basal markers CK5, 34βE12 and CD44. *KIFC1* expression was increased in basal type BC in some molecular classifications. These results suggest that *KIFC1* is involved in basal differentiation. The response of basal type BC to chemotherapy is controversial. A study by Choi et al. showed that p53-like type BC is resistant to neoadjuvant chemotherapy compared to basal and luminal type BC [18]. The study by Seiler et al., found that basal type BC benefitted more from neoadjuvant cisplatin-based chemotherapy than other BC subtypes [42]. The study by Taber et al., reported that basal type BC is associated with a poor response to cisplatin-based chemotherapy [24]. In the present study, Kaplan-Meier analysis showed that *KIFC1*-positive cases were associated with poor prognosis after cisplatin-based chemotherapy in BC. Although further studies are needed, *KIFC1* may be promising as a basal marker in BC.

A recent study found that DNA-damaging treatments induce *KIFC1* expression and *KIFC1*-dependent centrosome clustering [31]. Centrosome clustering contributes to chromosomal instability [43]. Indeed, our in silico analysis showed that *KIFC1* expression was increased in high genomic instability and genomically unstable BC subtypes. Loss of p53 promotes genomic instability in cancer cells [33]. In addition, in silico analysis showed that *KIFC1* alteration was associated with *TP53* alteration, and mRNA *KIFC1* expression was increased in patients with altered *TP53.* Immunohistochemistry showed that *KIFC1*-positive cases were associated with p53-positive cases. What is more, western blotting revealed that knockout of p53 induced *KIFC1* expression in BC cell lines. Taken together, these results suggest that the interaction between p53 and *KIFC1* may play an essential role in BC development and progression.

In the present study, immunohistochemistry showed that *KIFC1*-positive cases were associated with PD-L1 positive cases. Recent studies have shown that PD-L1 expression is increased in the basal/squamous subtype in BC [44,45]. In our study, *KIFC1* expression was also increased in the basal/squamous subtype. These results indicate that there may not be direct interaction between *KIFC1* and PD-L1. Although PD-L1 is used as a biomarker for ICI, the clinical utility is limited [36]. In our study, in silico analysis showed that high *KIFC1* expression was associated with favorable prognosis in BC patients treated with atezolizumab. Immunohistochemistry showed that *KIFC1* positive cases were associated with poor prognosis in BC patients treated with cisplatin-based chemotherapy. These results suggest that *KIFC1* may serve as a potential biomarker for drug selection.

As we mentioned above, *KIFC1* knockdown suppressed CD44 expression in BC cell lines. However, the mechanism is still unclear. A recent review reported that there is an inverse relationship between proliferation and differentiation [46]. Indeed, cisplatin reduces cell survival and induces differentiation of stem cells in breast cancer [47]. In our study, we showed that *KIFC1* was associated with cell proliferation signature. A previous study showed that *KIFC1* promotes cell proliferation [10]. These findings indicate that *KIFC1* knockdown may induce differentiation, which may help to explain why *KIFC1* knockdown suppressed CD44.

This study has some limitations. First, although we used siRNA to evaluate the function of *KIFC1* in BC, an overexpression model is needed to verify our findings. Second, immunohistochemistry showed that *KIFC1*-positive cases were associated with poor prognosis after cisplatin-based chemotherapy, but the number of samples was relatively small. Third, because *KIFC1*-positive cases were associated with favorable prognosis in BC treated with atezolizumab, in the future, we will analyze the prognostic value of *KIFC1* in immune therapy using immunohistochemistry. Fourth, although we showed that *KIFC1* knockdown promoted the sensitivity to cisplatin, the effect was not very dramatic. *KIFC1* was involved in bladder cancer proliferation, indicating that this involvement in proliferation may affect the sensitivity to cisplatin. In the future, we will validate the effect of *KIFC1* on cisplatin sensitivity in vivo analysis.

In conclusion, the present study revealed that high expression of *KIFC1* was associated with poor prognosis in BC, which was consistent with the findings from the public databases. *KIFC1* expression was increased in genomic instability and alteration of *TP53*. p53 knockout induced *KIFC1* expression, and *KIFC1* knockdown increased the sensitivity to cisplatin. Furthermore, prognosis was poor in the *KIFC1*-positive patients treated with cisplatin, whereas in patients treated with atezolizumab, *KIFC1* expression was associated with PD-L1 expression and a favorable prognosis. The data presented here highlight the great potential of *KIFC1* as a possible biomarker and therapeutic target in BC.

## Figures and Tables

**Figure 1 jcm-10-04837-f001:**
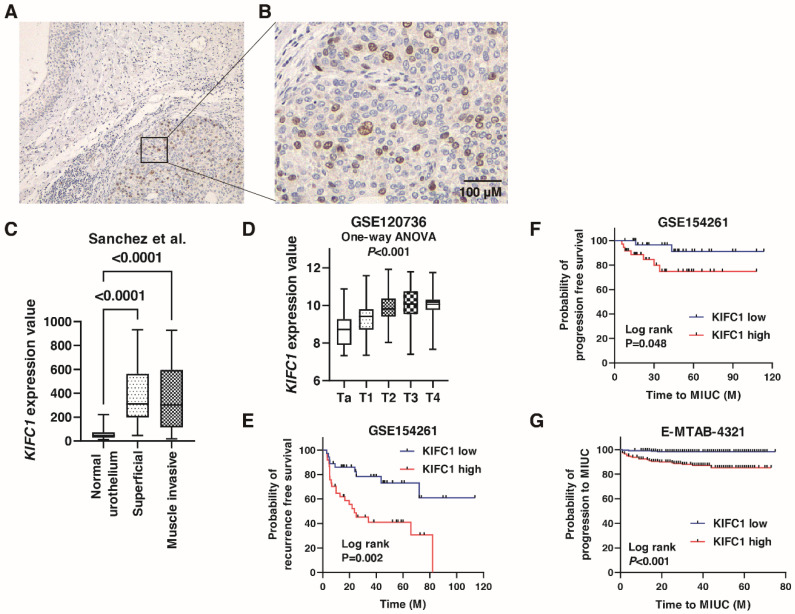
Expression of *KIFC1* in bladder cancer (BC). (**A**) Immunohistochemical staining of *KIFC1* in the non-neoplastic urothelium and BC. Original magnification: 100×. (**B**) Immunohistochemical staining of *KIFC1* in BC. Original magnification: 400×. (**C**) Box plot of *KIFC1* expression in normal urothelium, superficial BC, and muscle-invasive BC from the study by Sanchez et al. [21]. (**D**) Box plot of *KIFC1* expression according to T stage from the study GSE120736. (**E**) Kaplan-Meier plot of recurrence-free survival of T1 BC patients according to *KIFC1* expression from the study (GSE154261). (**F**,**G**) Kaplan-Meier plot of progression-free survival of T1 BC patients according to *KIFC1* expression after prostatectomy from the studies GSE154261 and E-MTAB-4321.

**Figure 2 jcm-10-04837-f002:**
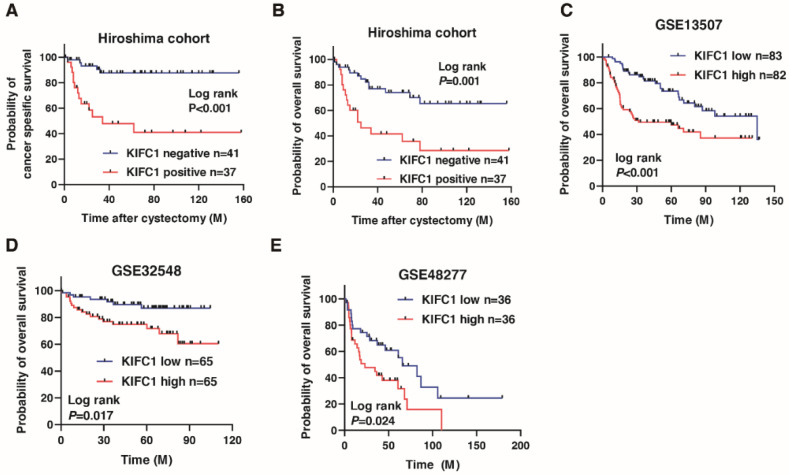
Prognostic value of *KIFC1* after cystectomy in bladder cancer (BC). (**A**–**E**) Kaplan-Meier plots of survival of BC patients after cystectomy according to *KIFC1* expression in the Hiroshima cohort and public databases (GSE13507, GSE32548, and GSE48277).

**Figure 3 jcm-10-04837-f003:**
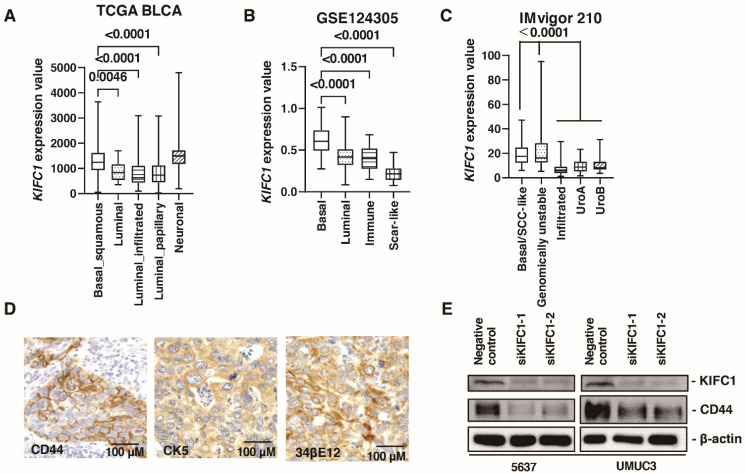
*KIFC1* is increased in basal type bladder cancer (BC). (**A**–**C**) Box plot of *KIFC1* expression by molecular classification in these studies: TCGA BLCA, GSE124305, IMvigor 210. (**D**) Representative immunohistochemical images of CD44, CK5 and 34βE12 expression in BC. Original magnification: 400×. (**E**) Western blotting of *KIFC1* and CD44 in 5637 and UMUC3 cells transfected with *KIFC1* or negative control siRNAs. β-Actin was used as a loading control.

**Figure 4 jcm-10-04837-f004:**
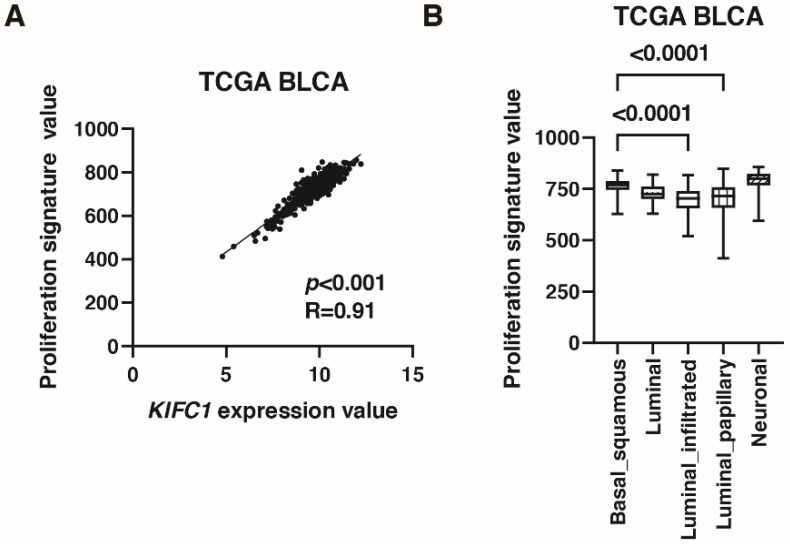
*KIFC1* is associated with proliferation in bladder cancer (BC). (**A**) The correlation between *KIFC1* expression and proliferation signature value in TCGA BLCA. Spearman’s correlation coefficients and *p*-values are indicated. (**B**) Box plot of proliferation signature value by molecular classification in TCGA BLCA.

**Figure 5 jcm-10-04837-f005:**
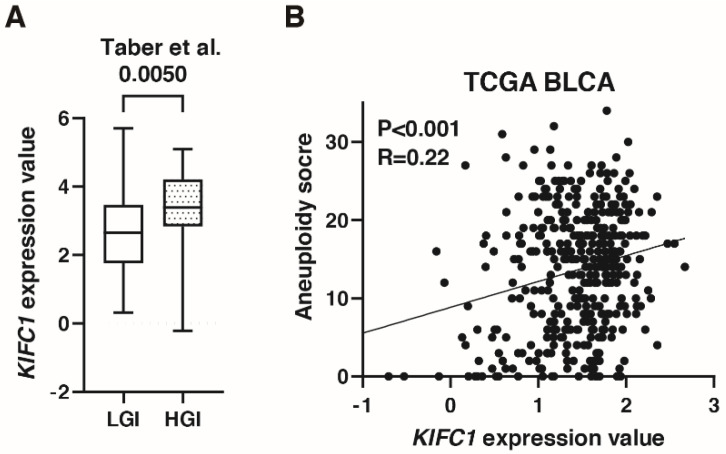
*KIFC1* is associated with genomic instability in bladder cancer (BC). (**A**) Box plot of *KIFC1* expression in low and high genomic instability groups in the study by Taber et al. [22] (**B**) The correlation between *KIFC1* expression and aneuploidy score in TCGA BLCA. Spearman’s correlation coefficients and *p*-values are indicated.

**Figure 6 jcm-10-04837-f006:**
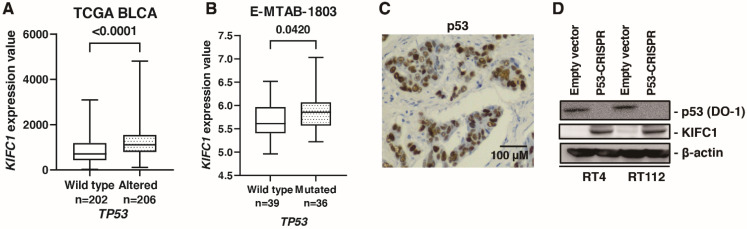
*KIFC1* is regulated by p53 in bladder cancer (BC). (**A**) Box plot of *KIFC1* expression in wild-type and altered *TP53* in TCGA BLCA. (**B**) Box plot of *KIFC1* expression in wild-type and mutated *TP53* in E-MTAB-1803. (**C**) Representative immunohistochemical images of p53 overexpression in BC. Original magnification: 400×. (**D**) Western blotting of p53 (DO-1) and *KIFC1* in RT4 and RT112 cells transfected with empty vector or p53-CRISPR. β-Actin was used as a loading control.

**Figure 7 jcm-10-04837-f007:**
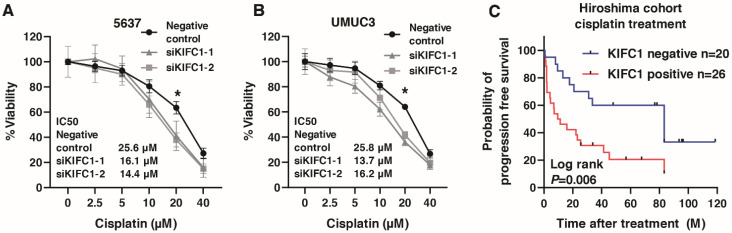
*KIFC1* is involved in cisplatin resistance in bladder cancer (BC). (**A**,**B**) The dose-dependent effects of cisplatin on the viability of 5637 and UMUC3 cells transfected with negative control siRNAs and *KIFC1* siRNAs. * *p* < 0.05. IC_50_ values are indicated. (**C**) Kaplan-Meier plot of survival of BC patients treated with cisplatin-based chemotherapy according to *KIFC1* expression in the Hiroshima cohort.

**Figure 8 jcm-10-04837-f008:**
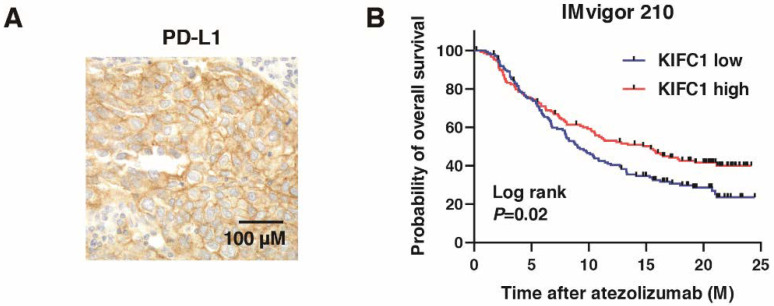
*KIFC1* expression is associated with PD-L1 expression and favorable prognosis after PD-L1 inhibition in bladder cancer (BC). (**A**) Representative immunohistochemical images of *KIFC1* and PD-L1 expression in BC. Original magnification: 400×. (**B**) Kaplan-Meier plot of survival of BC patients treated with atezolizumab according to *KIFC1* expression in the IMvigor 210 study.

**Table 1 jcm-10-04837-t001:** Relationship between *KIFC1* expression and clinicopathologic characteristics in the 78 bladder cancer from Hiroshima cohort.

	*KIFC1* Expression		*p*-Value ^a^
	Positive (*n* = 37) (%)	Negative (*n* = 41) (%)	
Age			
≤65 (*n* = 35)	14 (40%)	21 (60%)	N.S.
≥66 (*n* = 43)	23 (53%)	20 (47%)	
Gender			
Male (*n* = 63)	31 (49%)	32 (51%)	N.S.
Female (*n* = 15)	6 (40%)	9 (60%)	
Histological grade			
Low (*n* = 15)	7 (47%)	8 (53%)	N.S.
High (*n* = 63)	30 (48%)	33 (52%)	
pT status			
T1–2 (*n* = 54)	22 (41%)	32 (59%)	0.045
T3–4 (*n* = 24)	15 (63%)	9 (37%)	
Venous invasion			
Negative (*n* = 62)	27 (44%)	35 (56%)	N.S.
Positive (*n* = 16)	10 (63%)	6 (37%)	
Lymphatic invasion			
Negative (*n* = 46)	20 (43%)	26 (57%)	N.S.
Positive (*n* = 32)	17 (53%)	15 (47%)	
Lymph node metastasis			
Negative (*n* = 65)	28 (43%)	37 (57%)	0.043
Positive (*n* = 13)	9 (69%)	4 (31%)	

N.S. = not significant. ^a^
*p* values were calculated with Fisher’s exact test.

**Table 2 jcm-10-04837-t002:** Univariate and multivariate Cox regression analysis of overall survival in 78 bladder cancer.

	Univariate Analysis		Multivariate Analysis	
	HR (95% CI)	*p*-Value	HR (95% CI)	*p*-Value
Age (years)				
>65	1 (Ref.)		1 (Ref.)	
≤65	2.967 (1.329–7.510)	0.007	2.160 (0.904–5.283)	0.068
Venous invasion				
Negative	1 (Ref.)		1 (Ref.)	
Positive	3.052 (1.384–6.435)	0.007	1.227 (0.474–3.176)	0.672
Lymphatic invasion				
Negative	1 (Ref.)		1 (Ref.)	
Positive	3.286 (1.520–7.102)	0.003	1.906 (0.762–4.770)	0.168
pT stage				
pT1–2	1 (Ref.)		1 (Ref.)	
pT3–4	3.769 (1.798–7.901)	<0.001	1.476 (0.865–5.770)	0.367
pN stage				
Negative	1 (Ref.)		1 (Ref.)	
Positive	3.516 (1.494–8.277)	0.004	1.432 (0.535–3.824)	0.474
*KIFC1* expression				
Negative	1 (Ref.)		1 (Ref.)	
Positive	4.311 (1.903–9.766)	<0.001	3.121 (1.332–7.311)	0.009

HR: hazard ratio.

**Table 3 jcm-10-04837-t003:** Relationship between *KIFC1* expression and 34βE12, CK5, CD44 and PD-L1 in the 50 bladder cancer from Kure cohort.

	*KIFC1* Expression		*p*-Value ^a^
	Positive (*n* = 26) (%)	Negative (*n* = 24) (%)	
34βE12 expression			
Negative (*n* = 25)	9 (36%)	16 (64%)	0.022
Positive (*n* = 25)	17 (68%)	8 (32%)	
CK5 expression			
Negative (*n* = 25)	9 (36%)	16 (64%)	0.022
Positive (*n* = 25)	17 (68%)	8 (32%)	
CD44 expression			
Negative (*n* = 22)	7 (41%)	15 (59%)	0.011
Positive (*n* = 28)	19 (68%)	9 (32%)	
PD-L1 expression			
Negative (*n* = 39)	16 (41%)	23 (59%)	0.002
Positive (*n* = 11)	10 (91%)	1 (9%)	

^a^*p* values were calculated with Fisher’s exact test.

**Table 4 jcm-10-04837-t004:** Relationship between *KIFC1* expression and Ki-67 in the 58 bladder cancer from Hiroshima cohort.

	*KIFC1* Expression		*p*-Value ^a^
	Positive (*n* = 25) (%)	Negative (*n* = 33) (%)	
Ki-67 expression			
<20% (*n* = 26)	6 (23%)	20 (77%)	0.004
≥20% (*n* = 32)	19 (59%)	13 (41%)	

^a^*p* values were calculated with Fisher’s exact test.

**Table 5 jcm-10-04837-t005:** Relationship between *KIFC1* and *TP53* gene status in gene alterations in the TCGA BLCA.

	*TP53* Alteration		*p*-Value ^a^
	No (*n* = 211)	Yes (*n* = 197)	
*KIFC1* alteration			
No (*n* = 400)	210 (53%)	190 (47%)	0.018
Yes (*n* = 8)	1 (13%)	7 (87%)	

^a^*p* values calculated with Fisher’s exact test.

**Table 6 jcm-10-04837-t006:** Relationship between *KIFC1* expression and p53 in the 58 bladder cancer from Hiroshima cohort.

	*KIFC1* Expression		*p*-Value ^a^
	Positive (*n* = 29) (%)	Negative (*n* = 29) (%)	
p53 expression			
Wild-type pattern (*n* = 45)	26 (64%)	19 (36%)	0.024
Altered-type (OE + CA) (*n* = 13)	10 (77%)	3 (23%)	

^a^*p* values were calculated with Fisher’s exact test. OE: p53 overexpression, CA: p53 complete absence of expression.

**Table 7 jcm-10-04837-t007:** Clinicopathologic characteristics of 46 bladder cancer patients who were treated with cisplatin based chemotherapy.

Number of Cases	46
Gender	
female	10 (22%)
male	36 (78%)
Age (years)	42–82
Pathological T stage	
pTis	4 (9%)
pT1	11 (24%)
pT2	15 (33%)
pT3	10 (22%)
pT4	3 (6%)
Not evaluable	3 (6%)
Nodal metastasis	
Negative	29 (63%)
Positive	17 (37%)
Organ metastasis	
Negative	31 (67%)
Positive	15 (33%)
Chemotherapy setting	
Neo-adjuvant	16 (38%)
Adjuvant	30 (62%)
Response	
CR/PR	13 (28%)
SD/PD	29 (63%)
Not evaluable	4 (9%)

**Table 8 jcm-10-04837-t008:** Relationship between *KIFC1* expression and the response to cisplatin based chemotherapy in Hiroshima cohort.

	*KIFC1* Expression		*p*-Value ^a^
	Positive (*n* = 16) (%)	Negative (*n* = 26) (%)	
Response to cisplatin based chemotherapy			
CR/PR (*n* = 13)	6 (46%)	7 (54%)	N.S.
SD/PD (*n* = 29)	10 (34%)	19 (66%)	

^a^*p* values were calculated with Fisher’s exact test. N.S.: not significant.

**Table 9 jcm-10-04837-t009:** Relationship between *KIFC1* expression and the response to atezolizumab in IMvigor210 cohort.

	*KIFC1* Expression		*p*-Value ^a^
	High (*n* = 149) (%)	Low (*n* = 149) (%)	
Response to atezolizumab			
CR/PR (*n* = 68)	45 (66%)	23 (32%)	0.002
SD/PD (*n* = 230)	104 (45%)	126 (55%)	

CR: complete response, PR: partial reponse, SD: stable disease, PD: progression disease. ^a^
*p* values were calculated with Fisher’s exact test.

## Data Availability

All data generated or analyzed during this study are included either in this article or in the Appendix A.

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
