# Peer review of "KIFC1 Is Associated with Basal Type, Cisplatin Resistance, PD-L1 Expression and Poor Prognosis in Bladder Cancer"

_jcm, 2021, doi:10.3390/jcm10214837_

Round 1
Reviewer 1 Report
Sekino and colleagues have analyzed the expression of KIFC1 and the consequences of its knockdown in bladder cancer. The analyses are in general well performed, but the results must be contextualized better to be able to draw conclusions. Most critically, it should be established when KIFC1 simply acts as a measure of the proliferative activity of tumors versus when there is evidence that KIFC1 gene/protein itself is specifically important. For comments, see below.
1. In large transciptomic cohorts of bladder cancer, KIFC1 is one prototypical member of a large (>200 genes) coherent signature related to proliferation and cell division. This signature is related to but not a determinant of molecular subtype. The results shown in Figure 3A are very consistent with what would be seen for a proliferation gene signature in these cohorts. Proliferation is also often related to outcome, most simply because reflects each cancers malignant potential and that fast growing cancer tend to kill the patients faster in the absence of treatment. Since much of this work is presented as associations and effects specific for KIFC1, it must be tested to what extent KIFC1 is simply acting as a proxy for proliferation/mitotic index.
A necessary, major, addition to this work is therefore to show and test the association between KIFC1 expression and proliferation e.g. by IHC against Ki67 or another proliferation marker in the cohort. Furthermore, proliferation needs to be included as a covariate in outcome dependent analyses to show if it is KIFC1 specifically, or proliferation more generally that is driving differences in outcome. Finally, for every analysis where KIFC1 expression is associated to some categorical or continuous variable, (e.g. Fig3A-C, Fig4A-B, Fig5A-B, Fig 7A-C), it should also be reported if the difference could be explained by a similar difference in a proliferation signature/marker.
It would not invalidate the findings if some or all of the KIFC1-associations are due to differences in proliferation. The results still hold true but the interpretation differs: It should then be presented as associations to proliferation, a process for which KIFC1 may, in that case, serve as a marker.
If instead most of the presented analyses are found to be somewhat independent of proliferation, then the authors can still make the case for a specific importance of KIFC1 due to its independent function in centrosome formation during chromosome segregation. The work will in that case be much stronger.
2. It has been generally proposed that dividing epithelial cells differentiate less than its non-dividing counterparts (see e.g. PMID: 26825227). Given such an association, the result that CD44 is downregulated in 5637 and UMUC3 is also consistent with expression of KIFC1 in dividing cells. Knockdown of KIFC1 would then cause cells to stop dividing, which would allow them to differentiate, thereby expressing less CD44. Such an explanation would not require a mechanistic link between KIFC1 and CD44, and thus it needs to be ruled out for the claims in the manuscript to hold.
3. Immunohistochemical evaluation of p53 may need to be revised. If p53 IHC is used to study p53 activity/inactivation, a simple cutoff of 10% expression is not optimal. Non wild-type patterns, e.g. strong overexpression, but also complete absence or cytoplasmic staining indicate altered p53 (PMID:27840695).
4. The result in Figure 4D is not statistically significant and does not support to the following statement in the text: “In TCGA-BLCA, high KIFC1 expression tended to be associated with poor prognosis after platinum drug-based chemotherapy (Figure 6D).”
5. In the experiments shown in figure 6A-B sensitivity to cisplatin is higher after KIFC1 knockdown but the differences are not very dramatic. To demonstrate a specific effect of KIFC1 levels on cisplatin sensitivity it would be good to test if there is any difference in growth kinetics of these cell lines +/- KIFC1 knockdown also in the absence of cisplatin.
6. In Figure 6C, it would be interesting to see if there is an association to response in addition to survival in the Hiroshima cohort. If there is no link to response, but there is a link to survival it suggests a prognostic (As in Figure 2) rather than predictive role in this uncontrolled cohort comparison.
7. The analyses in the Imvigor cohort are weak.
a) Expression of many genes show some degree of association with immune-infiltration. In fact, any genes expressed specifically in cancer cells can be expected to have a (auto) correlation to immune-cell infiltration simply because, on average, the higher the proportion of immune cells in a tumor sample, the lower the proportion of cancer cells. The correlation plots, to me, are not enough to suggest any type of notable relationship between KIFC1 specifically and immune/TGF-beta status.
b) PD-L1 is higher in Basal/Squamous subtype (PMID 33365265, 29947424), which you argue is also the case for KIFC1. Thus you would expect a positive association between these markers simply by being expressed in the same subtype. Quite possibly, there is no specific link between the two proteins.
c) Given the results shown in Figure 2, the Kaplan Meier plot from the Imvigor cohort is consistent with a prognostic rather than a treatment predictive effect. Notably, this can be tested by checking for an association to treatment response. If KIFC1 is not associated with response but is associated with survival, it could be concluded that it is most likely a prognostic marker only, with no link to ICI-response.
8. I agree with the authors that “The response of basal type BC to chemotherapy is controversial.”. In light of this it would be interesting to know if staining with the basal markers also conferred a worse prognosis in the Hiroshima cohort, and if so, to what extent the KIFC1 remained a significant independent predictor after controlling for basal marker expression.
Author Response
Our responses to the reviewers' comments for jcm-1399035
Title:KIFC1 Is Associated with Basal Type, Cisplatin Resistance, PD-L1 Expression and Poor Prognosis in Bladder Cancer
Authors:Yohei Sekino, Quoc Thang Pham, Kohei Kobatake, Hiroyuki Kitano, Kenichiro Ikeda, Keisuke Goto, Tetsutaro Hayashi, Hikaru Nakahara, Kazuhiro Sentani, Naohide Oue, Wataru Yasui, Jun Teishima, Nobuyuki Hinata
Dear Editor and Reviewers,
Thank you for the reviewers’ comments concerning our manuscript entitled “KIFC1 Is Associated with Basal Type, Cisplatin Resistance, PD-L1 Expression and Poor Prognosis in Bladder Cancer”. Those comments are constructive and valuable for modifying and improving our manuscript. We have studied comments carefully and have fully addressed each comment. Revised portions are marked in red in the revised manuscript. I hope it is acceptable for publication in Journal of Clinical Medicine. The main corrections in the paper and the responses to the reviewer’s comments are as follows:
- The reviewer stated that the involvement of KIFC1 in proliferation may affect the findings from our study. Therefore, they proposed that we should analyze if KIFC1 is involved in cell proliferation.
Thank you for the valuable comments. We agree with the reviewer's comments. A previous study has shown that KIFC1 promotes cell proliferation in bladder cancer cell lines. We performed a signature analysis in TCGA BLCA cohort and immunohistochemistry of Ki-67. We found that KIFC1 expression was positively correlated with proliferation signature, and proliferation signature was increased in basal/ squamous type. Further, immunohistochemistry showed that KIFC1 positive cases were associated with Ki-67 positive cases. These results indicate that KIFC1 is involved in cell proliferation.
These statements were described in the revised results (page8, line 238-245) and table 4.
- The reviewer stated that there is an inverse relationship between proliferation and differentiation. Given that KIFC1 is involved in proliferation, KIFC1 knockdown may suppress CD44 expression without specific interaction between them.
Thank you for the essential comments. The reviewer's comment made us deeply understand this interaction. I added the description in the discussion. KIFC1 is involved in cell proliferation. So, KIFC1 knockdown may induce differentiation. However, in my experience, not all molecules related to cell proliferation affect CD44 expression. So, there may be some interaction between them.
These statements were described in the revised discussion (page14, line 396-403).
- The reviewer stated that our evaluation of p53 may be insufficient.
Thank you for the valuable comments. I reanalyzed the immunohistochemistry of p53 based on the study, which the reviewer recommended.
These statements were described in the revised discussion (page10, line 279-290).
- The reviewer pointed out that the result in Figure 6D is not statistically significant.
Thank you for pointing out our mistake. I removed figure 6D.
- The reviewer stated that the effect of KIFC1 knockdown on cisplatin sensitivity is small.
Thank you for the valuable comments. We agree with the reviewer's comments. Given that KIFC1 promotes cell proliferation, such phenomena may affect drug sensitivity. In the future, we will validate our findings by in Vivo analysis. We added the limitation in the discussion.
These statements were described in the revised discussion (page14-15, line 410-414).
- The reviewer proposed we should analyze the association between KIFC1 expression and the response to cisplatin-based chemotherapy.
Thank you for the valuable suggestion. The reviewer's comment made us deeply understand the difference between predictive value and prognostic value. We found that KIFC1 positive cases were not associated with the response to cisplatin-based chemotherapy. So, KIFC1 may be a prognostic marker for cisplatin-based chemotherapy.
These statements were described in the revised discussion (page11, line 306-307) and Table 8.
- a) The reviewer stated that KIFC1 expression is not specifically correlated with immune/TGF-beta status because cancer cells can be expected to have an (auto) correlation to immune-cell infiltration.
Thank you for the critical comments. We agree with the reviewer's comments. So, we removed the results of figure 7A-C. In the future, we will analyze this interaction in vitro.
- b) The reviewer stated that there may not be specific interaction between KIFC1 and PD-L1 because KIFC1 and PD-L1 are increased in basal/squamous type.
Thank you for the valuable comments. We agree with the reviewer's comments. We added the description in the revised discussion.
These statements were described in the revised discussion (page14, line 385-389).
- c) The reviewer proposed that we should analyze the association between KIFC1 expression and the response to atezolizumab in IMvigor 210 cohort.
Thank you for the valuable suggestion. We found that high KIFC1 expression was associated with the favorable response (CR/PR) to atezolizumab. KIFC1 may be a prognostic and predictive marker for atezolizumab treatment.
These statements were described in the revised discussion (page13, line 323-328) and Table 9.
- The reviewer proposed that we should analyze the association between some basal markers and the response to cisplatin-based chemotherapy and performed a multivariate Cox proportional hazard analysis.
Thank you for the valuable suggestion. We performed the immunohistochemistry of CK5/6 in the bladder cancer patients treated with cisplatin-based chemotherapy. However, the staining CK5/6 was not associated with the response and prognosis. We did not analyze other basal markers because the tissue samples from the bladder cancer patients treated with cisplatin-based chemotherapy are limited.
Reviewer 2 Report
The described data are interesting and supported by the experimental evidence; the paper is clear and well written.
The only part that doesn't seem well explained are the data reported in paragraph 3.5 describing the regulation of KIFC1 by p53 in BC. Therefore we ask to better rewrite this paragraph (lines 251-261).
Author Response
Our responses to the reviewers' comments for jcm-1399035
Title:KIFC1 Is Associated with Basal Type, Cisplatin Resistance, PD-L1 Expression and Poor Prognosis in Bladder Cancer
Authors:Yohei Sekino, Quoc Thang Pham, Kohei Kobatake, Hiroyuki Kitano, Kenichiro Ikeda, Keisuke Goto, Tetsutaro Hayashi, Hikaru Nakahara, Kazuhiro Sentani, Naohide Oue, Wataru Yasui, Jun Teishima, Nobuyuki Hinata
Dear Editor and Reviewers,
Thank you for the reviewers’ comments concerning our manuscript entitled “KIFC1 Is Associated with Basal Type, Cisplatin Resistance, PD-L1 Expression and Poor Prognosis in Bladder Cancer”. Those comments are constructive and valuable for modifying and improving our manuscript. We have studied comments carefully and have fully addressed each comment. Revised portions are marked in red in the revised manuscript. I hope it is acceptable for publication in Journal of Clinical Medicine. The main corrections in the paper and the responses to the reviewer’s comments are as follows:
- The reviewer proposed that we should rewrite the results regarding the association between KIFC1 and p53.
Thank you for the valuable suggestion. We revised the description regarding the association between KIFC1 and p53.
These statements were described in the revised results (page 10, line 279-290).
Round 2
Reviewer 1 Report
Remaining comments:
- In Figure 2, the authors should double-check the survival data. For example, in dataset GSE 32894, is the analysis really limited to the patients (very few) who recieved radical cystectomy? If not, why does the time axis state time from radical cystectomy. Check also that unique patients are only used once in the analyses (GSE 32548 is a subset of GSE 32549 which shares tumor samples from patients included in GSE 32894). Also confirm that the endoints for the KM curves OS/CSS is correct by comparing to each of the data sets original studies.
- Relating to the Kaplan-Meier curves. For a proliferation marker such as KIFC1, you wuold expect a much bigger difference in prognosis in cohorts with a wide stage-distribution. It would help if the authors characterized in figure 2 for each study the total n, and the stage distribution, at least indicating how many NMIBC/MIBCs that are included in the analyses.
- The authors should, in my opinion, consider when to use the term prognostic and predictive. I would refrain from using the term predictive if only survival outcome has been tested for a treated cohort. The reason for this is that a survival difference may still be completely independent of the treatment. The author may want to use "predictive" only when we can be sure that the effect must be due to the treatment - in my poinion this is when a) tests show association to response, or b) tests show association to survival in a study that includes a control group.
- The description of the p53 analysis is still lacking. What does positive/negative (Table 6) stand for in relation to wild-type or altered staining patterns? How many cases showed wild-type pattern, and how many showed each of the altered patterns (OE/CA/CY) ?
Author Response
Our responses to the reviewers' comments for jcm-1399035
Title:KIFC1 Is Associated with Basal Type, Cisplatin Resistance, PD-L1 Expression and Poor Prognosis in Bladder Cancer
Authors:Yohei Sekino, Quoc Thang Pham, Kohei Kobatake, Hiroyuki Kitano, Kenichiro Ikeda, Keisuke Goto, Tetsutaro Hayashi, Hikaru Nakahara, Kazuhiro Sentani, Naohide Oue, Wataru Yasui, Jun Teishima, Nobuyuki Hinata
Dear Editor and Reviewers,
Thank you for the reviewers’ comments concerning our manuscript entitled “KIFC1 Is Associated with Basal Type, Cisplatin Resistance, PD-L1 Expression and Poor Prognosis in Bladder Cancer”. Those comments are constructive and valuable for modifying and improving our manuscript. We have studied comments carefully and have fully addressed each comment. Revised portions are marked in red in the revised manuscript. I hope it is acceptable for publication in Journal of Clinical Medicine. The main corrections in the paper and the responses to the reviewer’s comments are as follows:
- The reviewer proposed that we should check the survival data from public databases.
- GSE32894
Thank you for the valuable suggestion. We did not know that GSE 32548 shares tumor samples from patients included in GSE 32894. So, I removed figure 2E.
- Time axis
We mistakenly thought that these samples were derived from radical cystectomy. We revised the time axis.
- Overall survival/ cancer specific survival
We are terribly sorry for our mistakes. We misunderstood the endpoints of each study. We revised figure 2.
- The reviewer proposed that we should show the detail clinical information of public databases.
Thank you for the valuable suggestion. I added the supplementary table showing the number of cases and stage distribution in these cohorts.
- The reviewer proposed that we should be careful when using term “prognostic” and “predictive”.
Thank you for the essential suggestion. In our study, KIFC1 expression was associated with the favorable response to atezolizumab and favorable prognosis after atezolizumab. So, we described that KIFC1 may be predictive and prognostic marker for atezolizumab treatment. I thought that this description was in line with your opinions. However, we are not sure that the effect is due to the treatment. So, we removed the term “predictive”.
- The reviewer proposed that we should rewrite the description of p53.
Thank you for the valuable suggestion. We reanalyzed the immunohistochemistry of p53. p53 overexpression was observed in 10 cases. p53 complete absence of expression was observed in 3 cases. Cytoplasmic staining of p53 expression was not observed. We rewrote the description of p53.
These statements were described in the revised results (page10, line 286-290) and table 6.